# A Novel Approach to Atomistic Molecular Dynamics Simulation of Phenolic Resins Using Symthons

**DOI:** 10.3390/polym12040926

**Published:** 2020-04-16

**Authors:** Matthew A. Bone, Terence Macquart, Ian Hamerton, Brendan J. Howlin

**Affiliations:** 1Department of Chemistry & Faculty of Engineering and Physical Sciences, University of Surrey, Guildford, Surrey GU2 7XH, UK; b.howlin@surrey.ac.uk; 2Bristol Composites Institute (ACCIS), Department of Aerospace Engineering, School of Civil, Aerospace, and Mechanical Engineering, University of Bristol, Queen’s Building, University Walk, Bristol BS8 1TR, UK; terence.macquart@bristol.ac.uk (T.M.); ian.hamerton@bristol.ac.uk (I.H.)

**Keywords:** material simulation, molecular dynamics, intermediate structures, phenolic resins, characterisation, symthons

## Abstract

Materials science is beginning to adopt computational simulation to eliminate laboratory trial and error campaigns—much like the pharmaceutical industry of 40 years ago. To further computational materials discovery, new methodology must be developed that enables rapid and accurate testing on accessible computational hardware. To this end, the authors utilise a novel methodology concept of intermediate molecules as a starting point, for which they propose the term ‘symthon’ (The term ‘Symthon’ is being used as a simulation equivalent of the synthon, popularised by Dr Stuart Warren in ‘Organic Synthesis: The Disconnection Approach’, OUP: Oxford, 1983.) rather than conventional monomers. The use of symthons eliminates the initial monomer bonding phase, reducing the number of iterations required in the simulation, thereby reducing the runtime. A novel approach to molecular dynamics, with an NVT (Canonical) ensemble and variable unit cell geometry, was used to generate structures with differing physical and thermal properties. Additional script methods were designed and tested, which enabled a high degree of cure in all sampled structures. This simulation has been trialled on large-scale atomistic models of phenolic resins, based on a range of stoichiometric ratios of formaldehyde and phenol. Density and glass transition temperature values were produced, and found to be in good agreement with empirical data and other simulated values in the literature. The runtime of the simulation was a key consideration in script design; cured models can be produced in under 24 h on modest hardware. The use of symthons has been shown as a viable methodology to reduce simulation runtime whilst generating accurate models.

## 1. Introduction

Materials development, much like drug development in the pharmaceutical industry, often relies on a systematic trial and error campaign of the design space for a given material system. Empirical trials can be time-consuming and costly, both financially and in terms of embodied energy, as many experiments must be done to explore all possible synthetic conditions. Increasingly, it is recognised that this approach is inherently unsustainable, since it also involves the production of waste material that must be disposed of safely. Computer modelling can be used to reduce empirical trial and error, and more quickly develop materials with increased functionality for bespoke applications. In the future, the aim of computational materials science is to conduct all trial and error testing virtually, using laboratory experiments to verify the integrity of a model and synthesise the material whose properties are optimal. Rather than spending time developing a single material, time is spent developing an accurate model that can predict a broad range of materials. Modern materials development is beginning to see the use of computer simulations for epoxy [1,2,3,4], carbon nanotube [5,6,7] and hybrid composite materials [8,9,10], to name but a few. High quality structures give rise to the accurate prediction of macroscopic physical properties, such as density, glass transition temperature (T_g_), and electrical conductivity [11,12]. The evaluation of these physical properties can be used to narrow down a broad design space to propose an optimised material, which can subsequently be tested empirically.

First produced by Leo H. Baekeland and patented in 1907, phenolic resins are considered the first wholly synthetic plastics, which were later commercialised as Bakelite [13]. Modern uses range from insulation foam in the construction industry to advanced aerospace applications such as heat shields employed by spacecraft on re-entry to the Earth’s atmosphere [14,15]. Phenolic resins are characterized by several key properties: high strength, low flammability, high char yield and low thermal conductivity [13,16]. These properties arise from the potential for crosslinking within the resin, but vary between resole and novolac resins. Resoles, the class of phenolic resin modelled in this work, have a formaldehyde (F):phenol (P) ratio of ≥1:1, up to a limit of 3:1 [17]. Excess formaldehyde facilitates extensive crosslinking between the chains. This crosslinking occurs via the formation of hydroxymethylphenol (HMP) intermediate structures, which can be seen in Figure 1. Novolacs have an F:P ratio of <1:1, and typically require curing agents, such as hexamethylene tetraamine (HMTA), to form solid materials at room temperature [18]. Resoles are the more commonly used form of phenolic resin, due to the higher strength and thermal resistance that they display compared to novolacs; they see continued use in high-temperature applications in aerospace [19]. For thermal protection within engineering applications, resole phenolics need to be bespoke to enable them to compete with other high-temperature thermosets, such as phenolic-triazine (PT) or phthalonitrile (PN) resins, or engineering thermoplastics such as polybenzimidazole (PBI), polybenzoxazole (PBO) or polyimides, in terms of performance and price.

There exist many different forms of structural modelling, the present work focusing solely on molecular dynamics (MD). MD uses the Born-Oppenheimer Approximation and classical mechanics to simplify modelling the movement of atoms under very short time intervals (typically 1 fs) for a time duration extending into microseconds [20]. Modelling can be visualised with atomistic rendering or coarse-grained approaches, representing groups of atoms as a single entity with combined properties. Simulations with high atom counts are generally considered more accurate; a unit cell with periodic boundary conditions is used to keep the simulation size practical yet scalable. Further to reducing computational demand, interaction cut-offs and energy forcefields are used to manage bonding and non-bonding interaction energies without the need to undertake ab initio calculations. Simulations can be influenced by macroscopic (bulk) properties with the implementation of thermostats/barostats to simulate the effects of temperature and pressure. Further details of these methods can be found in introductions to the topic by Allen and Tildesley [21] or Schneider et al. [22]. The design of the script to simulate the polymerisation of the system can affect the quality of the results, as well as the runtime. Common across many applications of computational simulation is the concept of static and dynamic approaches. Static approaches complete an action in its entirety, whilst dynamic approaches perform a subset of an action before continuing to carry out a major change such as structural movement. In MD, static approaches typically involve bonding every available bond; dynamic approaches bond a subset of the available bonds, before undergoing MD simulation, so as to avoid forming the structure too quickly. Monk et al. highlight in a sensitivity study that the dynamic methodology has a significant impact on the energy of bonded systems [23]. Dynamically cured systems tend to have lower potential energies, but have increased runtimes due to increased iterations of the curing script. 

The use of intermediates, or ‘symthons’, is an easy to implement design philosophy to reduce the runtime of simulations. Traditional MD simulations follow chemical mechanisms originating with real monomers. The first step in these simulations is bonding between the monomers. By contrast, originating the simulation with symthons, this initial bonding step is eliminated, and polymer chains can begin forming immediately. The HMPs outlined in Figure 1 reflect the second step of the mechanism in Figure 2, and form the basis of the symthons. This step has been proven empirically to pass through a quinone methide intermediate [24,25]. One of three products is formed based on whether the formaldehyde reacts at the *ortho-* or *para-* site, denoted as *o-o’*, *o-p’*, *p-p’* bonded phenols via a methylene bridge. The cure mechanism is known to be complex, and further side products have been identified empirically using spectral analysis, e.g., ^1^H and ^13^C nuclear magnetic resonance (NMR) or Fourier transform infrared (FTIR) spectroscopy, of phenolic resins [26]. These structures originate from the condensation of hydroxymethyl groups of two HMPs bonding together to form an ethylene bridge, or reacting at the hydroxide group.

A number of recent papers have aimed to generate phenolic resin structures; the comparison of their methods and results provide useful context to contrast the results of this work. All these works report density values which facilitate a direct comparison of the subtly different methods employed across the research. A key focus across all simulations of thermosets is enabling a high degree of crosslinking within short runtimes. Izumi et al. have demonstrated extensively the potential for MD within phenolic resin research [27,28,29]. They have shown that large (>200,000 atom, or 10,000 phenol molecules) MD simulations are suitable for modelling phenolic resins for studying inhomogeneity. However, their focus is on the static modelling of novolacs and using *pseudo*-reactions or united atom approaches which, being coarse grained, have less detail than an individual atomistic approach to bonding. Monk et al. have generated good dynamic phenolic models predicting density, T_g_ and Young’s modulus, which have been corroborated with empirical literature [23]. They also reviewed the effects of different modelling parameters on final structures, the results of which have been used to inform the design of the script in this work. Monk et al. have generated their structures from fixed length chains that have been crosslinked together, as opposed to monomers or a symthon. This is a simple method for generating crosslinked structures, although it limits the potential variability in the models, leaving the simulation prone to bias. Work by Li et al. utilises a novel mixed modelling technique for generating resole phenolic resin structures using quantum calculations, Monte Carlo and MD simulation originating from monomers [30]. Although they report model densities supported by empirical literature, their T_g_ results fall significantly below those produced empirically by Manfredi et al., with an F:P ratio of 2.0 simulating a T_g_ of 403 K, compared to 543 K from empirical resin derived by dynamic mechanical analysis [31].

The simulations performed in the present work aim to generate atomistic phenolic resin structures using a novel intermediate starting point (i.e., a symthon). Runtime is a focus of simulation design to enable rapid material testing, allowing the greater exploration of the novel material design space. Existing simulations in the literature, of both phenolics and other material systems, often develop a script that more closely represents the chemical mechanism of a curing polymer from starting reagents. By doing this, simulations have an increased computational demand, as the initial bonding of monomers is the primary step. Our novel methodology bypasses this initial step, significantly reducing runtime without impacting the accuracy of the structures produced. To examine further the effects of the simulation runtime, this simulation will be designed and tested on modest hardware in an effort to facilitate the more widespread use of materials modelling. The structures generated will be validated by comparing density and T_g_ to empirical and simulated results. Monk et al. show that density is intrinsically linked to Young’s Modulus and viscoelastic properties, therefore making it a vital parameter to be able to simulate accurately [23].

## 2. Materials and Methods

### 2.1. Software and Hardware

Modelling simulations were carried out using Materials Studio™ 6.1 by Accelerys™ Software Inc., with the additional packages Forcite Plus and Amorphous Cell. Programming and data analysis were undertaken on a desktop computer, with script writing in Perl for general flow control and Materials Script to interface with the model. To provide context for simulation runtime, simulations were managed by a 48 core Linux server of 4 AMD™ Opteron^®^ 6174 processors with 64 GB of RAM.

### 2.2. Initial Unit Cell Construction

Symthon (intermediate) structures were produced with *ortho*- and *para*- substitution sites, colored blue and purple respectively, which can be seen in Figure 3. This provided the script with the means to distinguish reaction sites from other parts of the structure. A cuboid periodic unit cell was constructed using the Amorphous Cell builder within Materials Studio, and symthons were packed into the cell to a target density of 1 g cm^−3^. Simulations were designed with 10,000 atoms per unit cell; the number of symthons in each simulation varied, but aimed to be the smallest value above 10,000 atoms possible. The F:P ratio could be controlled by varying the ratio of different mono- and di-substituted symthons. The cell dimensions varied with the F:P ratio, and ranged from 41–53 Å. The ratio of *ortho-* and *para-* sites was kept at 1:1, so as to not bias the bonding by favouring one site over another. Empirical literature is ambiguous over whether bonding is favored at the *ortho-* or *para-* sites [32,33,34]. Li et al. calculate that the activation energies of sites are comparable; to enable good comparison, this work follows the same evidence and does not bias the bonding of the structure [23]. The initial construction was repeated to create seven 10,000 atom structures, with F:P ratios ranging from 1:1 to 2:1.

### 2.3. Script Iterations

An average execution of the script will loop through 100 iterations of the processes outlined in Figure 4. All possible bonds within a 5 Å cut-off were narrowed down based on the closest contact between the formaldehyde moiety and the reaction site. To utilise a dynamic methodology, the array of possible bonds was shrunk to a maximum of 35 bonding pairs, which ensured the structure did not cure too quickly. These pairs were picked arbitrarily, and as other script components influence the number of bonds forming, the typical maximum number of bonds formed per script loop is around 25.

Following bond formation, as proposed by Izumi et al., bond angles are checked to determine whether the angle is realistic: bonds with unrealistic angles (< 90°) are broken [29]. Note that the bond angle used is the methylene bridge angle, rather than using the angle defined across the phenol rings before bonding. This check helped to reduce the potential energy of the finished structures. Following the formation of a successful bond, the hydrogen atom of the reaction site and the hydroxide group of the formaldehyde moiety are deleted from the system. This means the simulation reflects a perfectly dry phenolic resin (*N.B.* the polymerisation mechanism is a step growth, condensation mechanism that yields water molecules as a byproduct). As the rate of simulated cure decreased, two additional methods to promote curing and achieve a high final degree of cure were enabled. Formaldehyde moieties were slowly deleted from the system to free up potential reactive sites to enable crosslinking between chains and maintain the rate of cure. Furthermore, formaldehyde moieties were transferred from an occupied reactive site to neighbouring free reactive sites. This created further bonding by bringing formaldehyde moieties within the range of other free reactive sites, or by freeing up a reactive site that was in a range of other formaldehyde moieties that had not been moved. Both these methods were essential for F:P ratios > 1.5:1, as sites needed to be cleared of formaldehyde moieties to enable fully cured (> 99%) structures to be generated. In these cases, up to 10–15 formaldehyde moieties could be deleted per loop; deletion rates were ramped based on the number of moieties deleted in previous iterations, in order to maintain a steady deletion rate.

#### 2.3.1. Geometry Optimisation

The geometry optimisation is carried out with the COMPASS (Condensed-phase Optimized Molecular Potentials for Atomistic Simulation Studies) forcefield of pre-calculated bonding and non-bonding interaction terms. COMPASS was chosen over other popular fields such as DREIDING, as it has been specifically designed for condensed phase polymers such as thermoset networks [35]. Furthermore, COMPASS includes partial atomic charges, calculated by ab initio calculations, eliminating an additional step common in most simulations [2]. These charges are updated by Materials Studio as necessary as the structure cures. A combined steepest descent [36] and conjugate gradient method [37] was used to optimise the structures [38]. Steepest descents ran for a short (100 steps) optimisation to reduce the potential energy of the structure to something more feasible. The conjugate gradient method was found to be unsuccessful at high potential energies, as is typically seen initially, so was employed after steepest descents to achieve a better final optimisation. During the conjugate gradient stage, the geometry of the unit cell was allowed to change in order to reduce the volume of the cell, and thus the density of the system changed.

#### 2.3.2. Molecular Dynamics

This work made use of the NVT (Canonical) ensemble, which holds the number of atoms, the system volume and temperature as all constant. The NVT ensemble is feasibly employed in this simulation, as density is varied in the geometry optimisation stage; density is constant in the MD stage, as the volume is held constant. The temperature was held constant during dynamics simulations using the Nosé thermostat [39]. The Nosé algorithm gave the most control over temperature within the system when compared to the Andersen and Berendsen thermostats, which were also trialled. The initial system temperature was 298 K, which was gradually ramped between loops from 298 K to 398 K, based on the degree of cure of the system. The timestep of the simulation was 1 fs, and the initial simulation duration was 12 ps. This duration was increased to 20 ps as the structure became more cured, to allow the simulation more time to equilibrate. The velocity Verlet algorithm was used to integrate equations of motion across each timestep [40]. Motion groups were not kept rigid throughout this process, to facilitate random movement within the simulation. Once the structure reached the cut-off degree of cure, the loop was closed with a final 20 ps dynamics simulation at 298 K, to equilibrate the structure to room temperature for comparison with literature data.

### 2.4. T_g_ Simulation

To identify the T_g_ of the final cured structures, the Temperature Cycle protocol from the Amorphous Cell module was used. Structures were elevated from an initial temperature of 300 K to 600 K. Finalised structures with F:P ratios of 1.2:1 and 1.5:1 were simulated. The protocol works by gradually raising the temperature of the system in 10 K increments, followed by simulating atom dynamics, allowing the volume of the system to change. Each temperature increment was modelled five times, and a plot with two fitted lines was produced. The T_g_ can be found by observing the temperature at which the density of the system decreases considerably—the point at which the two fitted lines intersect [41]. This corresponds to an increase in volume, as the system transforms from a rigid structure to a more flexible, elastomeric one.

## 3. Results and Discussion

### 3.1. Simulation Outcomes

The results of seven cured 10,000-atom structures across a range of F:P ratios derived from the method herein are displayed in Table 1. The density results are reproduced graphically in Figure 5 for ease of comparison with results from other groups. As can be seen, there is a broad range of densities quoted within the literature [17,42]. In comparison to the results of Manfredi et al., these results follow the trend that peak density lies somewhere around an F:P ratio of 1.2:1 [31]. Equally, the results show good comparison to those reported by Li et al., the closest simulation in literature, for F:P ratios ≤ 1.5:1—however, the reliability of these results is brought into question above 1.5:1. A greater than 1.5:1 F:P ratio marks a change in design philosophy for the simulation, as the formaldehyde component deletion/movement becomes essential to achieve the high degree of cure seen in the results of Table 1. The positive potential energies of > 1.5:1 F:P ratio models are thought to contribute to the non-conforming density results. 1.6:1 and 1.8:1 F:P ratio structures have similar densities but vastly different potential energies. The 2:1 F:P model has density in agreement with the literature, but high potential energy. There is no clear trend in the discrepancy between the results to suggest a definitive cause for this error. One thought is that formaldehyde deletion and movement methods are causing the simulation to run longer, which hampers structure relaxation methods. Various methods have been trialled to produce more relaxed cured structures, using longer duration simulations and/or shorter timesteps, but all significantly increased the runtime without meaningfully improving potential energy. Additional simulation and refinement of the script procedure is required to fully establish the cause; however, the models in the ideal F:P ratio range defined by the literature are conforming, which gives credence to the model design. 

Simulated values of T_g_ were derived for the 1.2:1 and 1.5:1 F:P ratio cured models, and achieved values of 515 K and 526 K, respectively. A plot of density as a function of temperature for the 1.5:1 model can be seen in Figure 6. A clear decrease in density at 526 K is observed, corresponding to an increase in volume as the structure becomes more flexible. Comparing these results to literature provides good validation for this simulation design. Manfredi et al. reported T_g_ values of 506 K and 530 K for 1.2:1 and 1.5:1 F:P ratios, respectively [31]. In comparison to other simulated literature, the simulation in this work shows a clear improvement, e.g., the comparable simulation produced by Li et al., which resulted in a T_g_ of 412 K for a 1.5:1 F:P ratio model [30].

### 3.2. Evaluation of Novel Modelling Concepts

The runtimes for these simulations ranged from 12 h for 1:1 F:P ratio structures to 23 h for 2:1 ratio models. The discrepancy in timing originates from the increased reliance on formaldehyde moiety deletion and movement in the higher F:P ratio models. Using these methods to achieve a high degree of cure led to more script iterations, and thus longer runtimes, in higher F:P ratio models. There is room for further refinement of these methods to ensure that deletion/movement is high enough to promote fast simulations without impacting the final degree of cure. Considering the results presented in Table 1, it may be suggested that these methods were too severe for the 1.5:1 and 1.6:1 models, as they fall further below their theoretical maximum degree of cure than other models.

In early script design, the NPT (Isothermal-isobaric) ensemble was used, as it allows the density of the structure to vary during dynamics simulations. However, concern can be found in the literature, as well as in discussion with our group, over the ability of the NPT ensemble to meet its ensemble averages [43,44]. Equally, results produced by the NPT ensemble in this simulation had density values ranging from 1.02–1.10 g cm^−3^, far lower than seen in most empirical literature. This was the case across the range of F:P ratios trialled. Varying the barostat, attempting the Andersen, Berendsen and Parrinello Rahman algorithms [45,46,47], had a limited effect on the results produced. Materials Studio uses an NPT ensemble developed by Martyna et al. in 1994, and subsequent iterations have attempted to improve upon this [44,48,49,50]. In order to switch to the more reliable NVT (Canonical) ensemble, an alternative means of altering volume across the cure loops needed to be adopted. During the geometry optimisation phase, both the model structure and the unit cell dimensions were optimised, enabling the unit cell to change volume, and altering the subsequent density of the total structure. This novel methodology has been shown to produce acceptable structures in this instance, as seen by comparison with empirical results herein.

The use of symthons has been shown to be capable of producing phenolic structures that fall in line with empirical and simulated literature; their use raises questions regarding static and dynamic bonding within molecular dynamics (MD). Using symthons as a starting point is similar to a single large-scale static cure of phenol and formaldehyde molecules in a conventional phenolic resin simulation. Based on this concept, the evidence herein shows that there is scope to vary static and dynamic bonding in MD simulations. Early stage bonding, if not employing symthons and instead originating with monomers, can be modelled statically or with looser dynamic bonding parameters. As the structure cures, bonding should become more dynamic—potentially scaling down bonds per cure cycle gradually right to the end of the simulation.

## 4. Conclusions

The fundamental aim of this work was to produce an accurate (i.e., chemically representative) large-scale model originating from an intermediate structure (a symthon). Further to this, a new modelling methodology was trialled using the NVT (Canonical) ensemble and unit cell dimension optimisation during the conventional geometry optimisation stage. These methods have proven successful when applied to phenolic resins, as models have well-reproduced literature values arising from both empirical and simulated experiments. Low runtimes have been achieved for these simulations, with fully cured models being generated within 12–23 h on readily accessible hardware. Moreover, there is scope for a further reduction in runtime by exploring how novel formaldehyde deletion/movement methodology affects the final structure. Symthons have successfully been shown to be a viable methodology for simplifying simulations, ultimately reducing runtimes. Their use in other polymer systems warrants additional investigation. Equally, research into the use of symthons to promote ease of simulation, rather than simulations based on replication of chemical mechanisms, would be a valuable addition to the concept. Further simulation refinement is required to reduce the final potential energy of structures with an F:P ratio >1.5:1, as this currently limits the scope of this simulation.

## Figures and Tables

**Figure 1 polymers-12-00926-f001:**
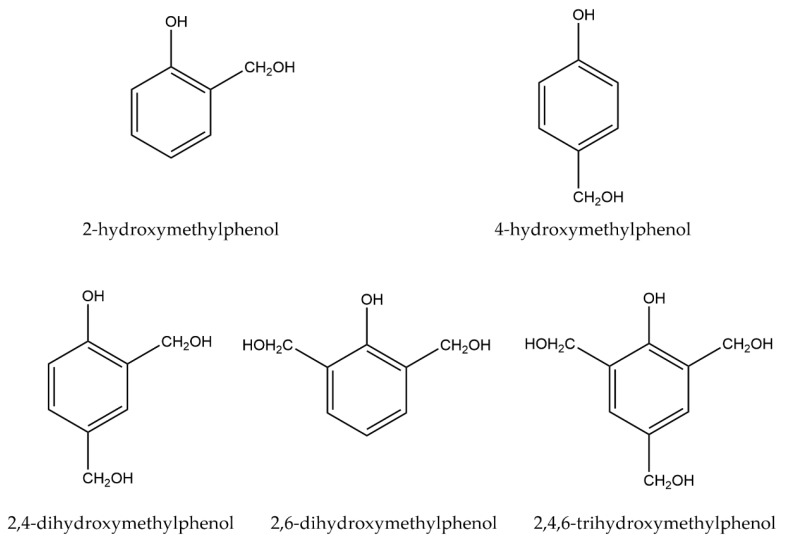
Potential HMPs formed from mono-, di- or tri-substitution of an initial phenol molecule with formaldehyde.

**Figure 2 polymers-12-00926-f002:**
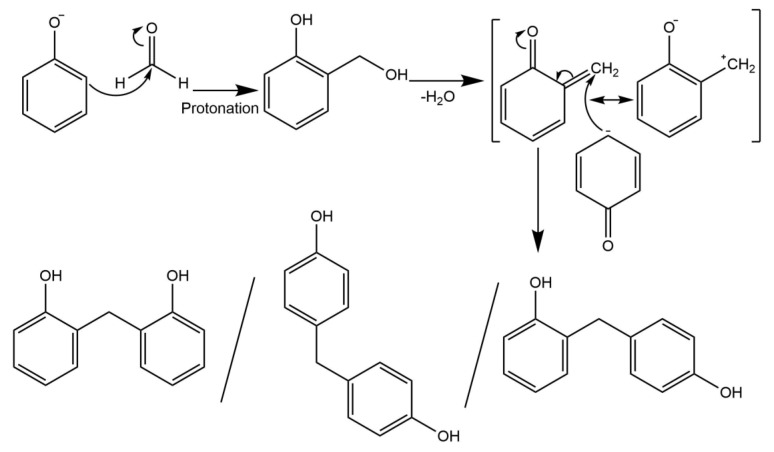
Polycondensation mechanism phenol and formaldehyde reacting via a quinone methide intermediate to form a methylene bridge. Possible products from left to right: *o-o’*, *p-p’*, *o-p’*.

**Figure 3 polymers-12-00926-f003:**
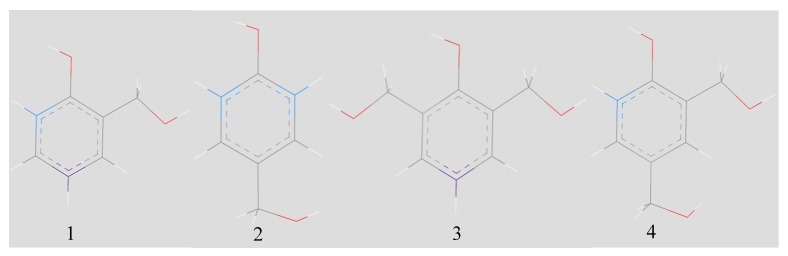
Mono-substituted (1 and 2) di-substituted (3 and 4) symthon structures used to populate unit cells to achieve different F:P ratios. *Ortho* (blue) and *para* (purple) sites are colored to give them a unique identifier for which the script can search.

**Figure 4 polymers-12-00926-f004:**
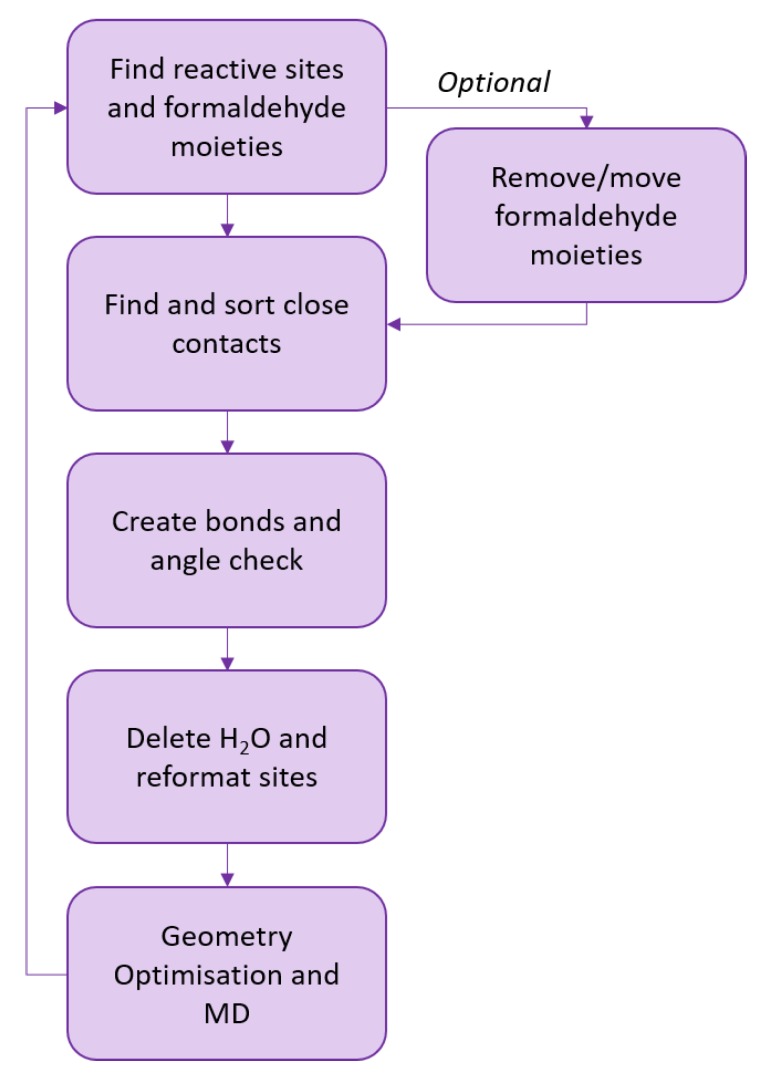
A simplified overview of the script design. The optional formaldehyde moiety deletion/movement methods would be activated when the relative degree of cure between iterations fell too low. This was essential for F:P ratios > 1.5:1, where reactive sites need to be cleared to achieve high degrees of cure.

**Figure 5 polymers-12-00926-f005:**
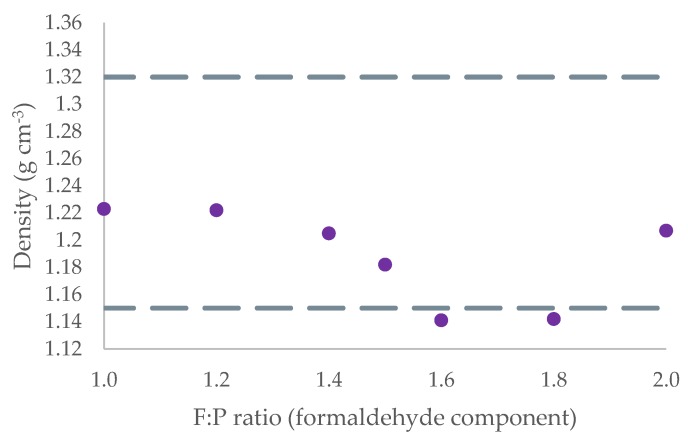
Graphical representation of density as a function of F:P ratio for cured 10,000 atom structures. Grey dashed lines indicate the upper and lower bound densities found within the literature [17,42].

**Figure 6 polymers-12-00926-f006:**
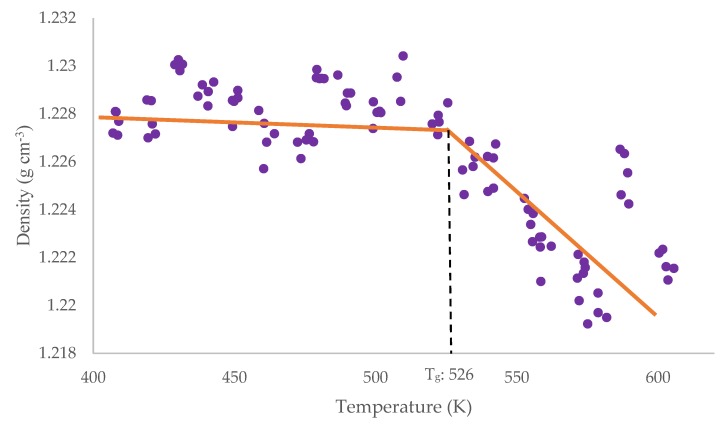
Density against temperature plot for a cured 1.5:1 F:P ratio model. T_g_ is observed at the gradient change as 526 K.

**Table 1 polymers-12-00926-t001:** Results of seven cured 10,000-atom structures of varying F:P ratios.

F:P Ratio ^1^	Density/g cm^−3^	Degree of Cure/%	Maximum Theoretical Degree of Cure/%	Initial Atom Count	Final Atom Count	Final Potential Energy/kcal mol^−1^
1.0	1.223	66.1	66.7	10,013	8261	−13,367
1.2	1.222	79.1	80.0	10,057	8039	−11,051
1.4	1.205	91.6	93.3	10,044	7790	−5107
1.5	1.182	96.8	100.0	10,032	7659	−2975
1.6	1.141	97.3	100.0	10,012	7489	2491
1.8	1.142	99.5	100.0	10,020	7220	43,785
2.0	1.207	99.2	100.0	10,017	6947	41,185

^1^ F:P ratio refers to the molar ratio of formaldehyde in the initial unit cell.

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
