# Peer review of "A Novel Approach to Atomistic Molecular Dynamics Simulation of Phenolic Resins Using Symthons"

_polymers, 2020, doi:10.3390/polym12040926_

Round 1
Reviewer 1 Report
In their manuscript “A Novel Approach to Atomistic Molecular Dynamics Simulation of Phenolic Resins Using Symthons” Bone et al. proposed a new approach to model phenolic resins using molecular dynamics simulations. The authors detailed the steps of the approach clearly. I outlined the suggestions and questions below for improving the clarity of the manuscript. I would support reconsidering the paper for publication in Polymers after minor revision.
(1) Do the authors calculate the radial distribution plots both in liquid (before polymerisation) and in resin (after polymerisation) for each F:P ratio? Providing these plots would be helpful to reader to understand how this ratio affects the structuring in the liquid?
(2) Have the authors updated the partial charges of the reacted atoms?
(3) Have the authors checked the average bond length of the newly formed bonds and average angle for each F:P ratio (please check Soft Matter 12 (2016) 2453-2464)? This may help understand why final potential energies appeared to be very big, especially for the samples F:P ratio of 1.8 and 2.0.
(4) Can the author provide the heating/cooling rate during the calculation of glass transition temperature (Tg) in the manuscript? Due to the discrepancy between the cooling rates implemented in the simulations and used in the experiments requires use of a correction term. Authors can check Polymer 191 (2020) 122253 for the reference of the correction term and its use in the crosslinked resins.
The approach the authors have developed in this work will be very helpful to readers. However, using freely available software (e.g. LAMMPS) and force-fields (e.g. DREIDING) will help increase the number of researchers who will use this approach in their research.
Reviewer 2 Report
Please see attached file.

Reviewer 3 Report
Authors propose to built atomistic models by using intermediate structures, so-called symthons, to speed up the equilibration step. I wonder why there is something novel about it. It is still possible that some people use a less plausible approach to build their models, but I haven't heard about before. Authors mention in their introduction that it might still be that in their area of research highly unsophisticated commercial software is in use. For such readers the present contribution could be eye-opening. In general, however, I believe that what is proposed here is trivial, and is routinely done by most researchers, especially by all those who build their model by themselves.
It should be clearly mentioned why the present approach helps to reduce the simulation time after the initial structure had been formed. Does it at all, and if so, why?
During the generation process, are the sympthons rigid bodies?
Authors mention that the NVT ensemble is more reliable, but how would they find the density of their system using NVT? This statement is highly confusing.
How did authors obtain Figure 6 without using NPT?
Please explain in more detail why for structures with a F:P ratio > 1.5:1 the energy comes out to be too large. What exactly is the problem?
Simulation times should be mentioned explicitely and compared with estimated for other methods, ie. other methods should be listed and compared in a more quantitative fashion.
Round 2
Reviewer 2 Report
The paper may be published in its present form.
Reviewer 3 Report
Authors took into account the recommendations with their revised manuscript.